# Digital governance, anti-corruption and political stability: An empirical study using cross-national panel data

Yaxing Zhao, Zhizhou Du ⓘ *

School of Public Administration, Yanshan University, Qinhuangdao, China

* duzhizhou@126.com

## Abstract

Digital governance has emerged as a critical domain for national development and international cooperation. This study investigates the impact of digital governance on political stability through both theoretical and empirical analyses. First, we establish a theoretical framework to examine the effects of digital governance on political stability and the role that anti-corruption plays in it. Using panel data from 112 countries during 2014–2023, we then examined the quantitative relationship between digital governance and political stability. The results show that digital governance significantly enhances political stability, mediated by anti-corruption efforts. Additionally, the impact of digital governance exhibits heterogeneity across different levels of economic development. Based on these insights, we provide policy recommendations and future research directions to leverage digital governance for enhancing political stability.

## 1. Introduction

In recent years, global political stability has faced mounting challenges. For example, escalating geopolitical conflicts, weakening multilateralism causing sudden problems in the global governance system, and technology and information warfare undermining trust and cooperation among countries. These phenomena have collectively heightened uncertainty in international political stability. However, political stability remains the cornerstone of national and global development. It is not only the foundation for economic growth, social harmony, and livelihood protection but also a vital prerequisite for maintaining peace and security worldwide. In order to cope with global challenges, prevent escalation, and examine the chain reactions of conflicts, it is urgently necessary to enhance global political stability. According to Hurwitz [1], political stability can be defined based on five aspects: (1) Stability refers to the absence of violence; (2) Stability refers to the durability of a government's lifespan;

**Data availability statement:** All relevant data are within the paper and its Supporting Information files.

**Funding:** The author(s) received no specific funding for this work.

**Competing interests:** The authors have declared that no competing interests exist.

(3) The stability of its existence as a legitimate constitutional order; (4) Stability of the structure without change; (5) Stability as a multi-faceted social attribute.

Digital governance aims to utilize digital technologies and data resources to govern and manage various fields such as the economy, society, and government. Its application scope is constantly expanding, and an increasing number of countries are incorporating it into their governance frameworks. For instance, Uruguay has launched the firma.gub.uy website, offering advanced electronic signature services from multiple providers registered with the electronic certification department. This is convenient for individuals and enterprises to use and verify, aiming to support all online cross-border transactions, save users' time and money, simplify administrative processes, reduce transaction obstacles, and enhance the productivity and competitiveness of enterprises; also, Singapore's application of artificial intelligence in public services has enhanced service efficiency in the healthcare and transportation sectors [2]. Existing research have shown that the government's use of digital technology can effectively maintain political stability [3]. For instance, website construction is an essential part of national digital governance strategies and gradually permeates national governance systems into a global trend [4,5]. Beyond improving the operational efficiency and optimizing the business environment [6], e-government has shown potential in curbing corruption and crime [7]. However, a study of member countries of the Association of Southeast Asian Nations found that while the penetration of e-governance has the potential to improve good governance, it may itself expand corrupt practices in some ways [8].

Digital technology is also a instrumental tool for countries to fight corruption. For example, Russia has established competitive procurement procedures in electronic form on electronic platforms, allowing the use of artificial intelligence technology to detect economic anomalies and signs of formal competition in government contracts. Meanwhile, since January 1, 2021, cryptocurrencies have been regarded by many countries as objects of anti-corruption monitoring [9]. Furthermore, Seiam and Salman [10] used panel data from 110 countries between 2003 and 2021 to examine the importance of e-government in fighting corruption and improving transparency, and the results show that e-government plays a leading role in fighting corruption and improving transparency. E-government is one of the decisive factors in fighting corruption, but it still needs to develop to a certain level to effectively combat corruption [11]. In addition, according to the comparative results of e-government and corruption levels, the higher the level of e-government development, the lower the level of corruption [12]. According to Bussell [13], the level of existing corruption in a country can be a stable predictor of the ultimate effectiveness of e-government in that country.

The existing research on political stability, digital governance, and corruption has achieved many theoretical and practical results, but there are still three limitations in the existing research. First, researchers pay little attention to the direct impact of digital governance on political stability, and most of the research objects are from developing countries. However, political stability is a global issue, and digital governance is a prevalent problem for both developing and developed countries. Second, many researchers neglected the long-chain effect of digital governance on political

stability. Third, the mediating role of anti-corruption lacks empirical quantification. Therefore, the research questions of this study are as follows: Does digital governance have a direct effect (influence) on political stability from a global perspective? How does digital governance indirectly improve political stability by mitigating corruption? Is there any heterogeneity in the above mechanisms at different economic levels? To solve the problems mentioned above, this research conducted an extensive study on how digital governance affects political stability, and how digital governance indirectly improves the degree of political stability by mitigating corruption through the use of econometric models, which is based on cross-country panel data from 112 countries during 2014–2023, intending to evaluate the transmission mechanism of digital governance to political stability. In this way, the purpose of this research is to enrich theoretical discourse on the relationship between digital governance, anti-corruption, and political stability; Provide a basis for developing technology-enabled anti-corruption policies and enhancing political stability; Fill the existing literature on the factors affecting political stability of the empirical evidence gap.

The remainder of this article is organized as follows: Section 2 delivers a literature review and research hypothesis. In section 3, model selection and detailed data description are given. Section 4 provides empirical research results and discussion. In section 5, conclusions, policy implications, and future research directions are discussed.

## 2. References and assumptions

### 2.1 The direct impact of digital governance on political stability

The World Bank defines e-governance as the use of information technology by government agencies. With the continuous development of e-governance, its conceptualization has been evolved more and more significantly. According to Sharma [14]: "e-governance may be defined as the delivery of government services and information to the public by using electronic." From a contemporary perspective, e-governance is seen as a broader term that encompasses not only a range of relationships and networks within government but also information and communication technologies [15]. Furthermore, digital governance is a government governance strategy, that is, the advanced use of information and communication technology to improve organizational performance [4]. Erkut [16] believes that e-government or digital government is a structural component of e-governance: E-government is a practical structure in which the government uses information and communication technology to realize the interaction between government and citizens, government to government, government to enterprise, and other levels; E-governance is a process involving multiple stakeholders to determine the direction, form, and extent of Internet activities. Synthesizing these perspectives, digital governance means that the government uses digital technology as a means of governance to integrate various factors such as technology, management, system, and subject participation to achieve effective governance and sustainable development in various fields of society.

The relationship between good governance and political stability has been well-established in the literature. According to Martins [17]: "Good Governance is defined as the processes and structures that guide political and socio-economic relationships in a way perceived as positive and fruitful for the society." There comes the question: how does digital governance promote good governance? The logic behind this is that digital governance forces people to think deeply about the role and impact of technology in every activity of the organization, it can prevent people from making mistakes, improve the effectiveness of decision-making, and improve accountability by recording every interaction [18]. Hu and Zhang [19] have accomplished a field investigation in China from 2019 to 2022 and found that e-governance, as an important component of digital governance, also plays a positive role in maintaining political stability, in the meantime, China has been using e-governance to strengthen the political stability and the people satisfaction. Studies on ASEAN countries show that the degree of political stability depends on the political and government stability and peace, and citizen happiness has a significant impact on political stability, also, the adoption of e-government will improve people's happiness level [20]. Case studies from transitional contexts, such as Mohamed's analysis [21] on political instability and e-government in Libya, further demonstrates that e-government as a tool for change can play a strategic role in bringing political stability and citizen

satisfaction. Moreover, the promotion and development of e-government can create an intelligent governance model, and promote the government transformation to the miniaturization direction, ethics, accountability, agility, and transparency [22]. In sum, digital governance helps to improve governance, and good governance plays a positive role in political stability. Therefore, the higher the level of digital governance, the stronger the political stability. The first hypothesis of this paper is proposed:

H1: Digital governance has a direct impact on the country's political stability.

## 2.2 The indirect impact of digital governance on political stability

Corruption represents a persistent global challenge with profound socio-political consequences. Different scholars and institutions have different definitions of corruption. The World Bank defines corruption as the "use of public for private gain" [23], emphasizing the nature of the behavior of using public resources for private gain. Mungiu-Pippidi and Hartmann [24] argue that when corruption becomes a systemic problem, it will evolve into a social norm, and individuals will adapt and not resist, which becomes a social dilemma. Shabbir et al. [25] define corruption as "the unfair and illegal activities of those in power." Existing research shows that corruption has profound negative effects on political stability and economic development. The International Monetary Fund [26] points out that corruption impedes economic growth and, in turn, contributes to political instability. To be precise, economic disruption can lead to social conflicts and popular discontent, further affecting political stability. More than that, reduced investment and slower economic growth will affect social stability and development. Sokim et al. [3] argue that corruption is one of the most damaging consequences of poor governance and can hinder sustainable investment and economic growth. Anderson and Tverdova [27] found that corruption undermines trust in public servants and worsens voters' perceptions of the political system through a survey of different countries. It can be seen that the lack of trust will weaken the legitimacy and authority of the government and lead to political instability. In addition, the study of Farzanegan and Witthuhn [28], using panel data for more than 100 countries over the period 1984–2012, found that corruption becomes politically destabilizing when the youth bulge exceeds a critical level of about 20 percent. Moreover, corruption also increases the risk of ethnic civil war. Neudorfer and Theuerkauf [29] show that the impact of corruption on the risk of ethnic civil war is positive, and ethnic civil war is a major factor in government instability, which means conflict and instability can seriously undermine political stability.

In recent years, digital governance has been seen as an effective tool to reduce corruption. Hanisch and Goldsby [30] propose that digital governance can effectively reduce corruption by improving efficiency and transparency. For example, electronic service delivery can reduce interactions with officials, speed up decision-making, and minimize human error [31]. The transparency of public institutions has a significant positive impact on a country's perceived level of corruption [32,33]. The case of Ukraine is further proof of the effectiveness of digital governance. Research by Olha [7] shows that the Ukrainian government has actively promoted e-governance through legislative measures, which has significantly improved the transparency of public services. At the same time, several studies have demonstrated the positive impact of government efforts to reduce corruption through digital technologies. The research of Kozerivska et al. [34] further supports this view that digitization reduces the risk of intervention and abuse by providing transparency and process automation, thereby helping to fight corruption and ensure the economic security of the country. Specifically, digital technologies such as information and communication technologies (ICT) can help to monitor the behavior of public officials by promoting openness and transparency, thereby reducing corruption [35]. Hayitov [36] also points out that the use of ICT in public administration, business, education, and healthcare can improve the transparency of service delivery and thus decrease the level of corruption. In addition, e-government plays a vital role in combating corruption [37,38]. Existing studies have shown that the development level of e-government is negatively correlated with the corruption level, among which the environment sub-index, usage sub-index, and telecommunications infrastructure sub-index are the key factors affecting the level of corruption [39,40]. It can be seen that existing studies have elaborated the definition of corruption

from different angles, and a large number of empirical studies have shown that corruption has a multi-faceted destructive effect on political stability, including economic growth obstruction, lack of trust, poor governance, and increased risk of conflict. Digital governance has shown remarkable results in reducing corruption by increasing transparency, automating processes, and monitoring behavior. It can effectively decrease corruption levels, increase public trust in government, and promote sustainable economic development, thereby enhancing political stability. In summary, the second hypothesis of this paper is proposed:

H2: Digital governance impacts political stability by changing the strength of anti-corruption.

## 3. Data and method

### 3.1 Model setting

**3.1.1 Benchmark regression model.** In order to verify H1 and explore whether the digital governance level can affect the degree of political stability. The benchmark regression model in this study is set as follows:

$$ln\ PS_{it} = \beta_0 + \beta_1 DG_{it} + \beta_2 \sum control_{it} + \varnothing_i + \varepsilon_{it} \tag{1}$$

Although the individual fixed effects model effectively addresses a major source of omitted variable bias by eliminating individual characteristics that do not change over time, such as a country's historical culture and geographical environment, it cannot automatically guarantee that the error terms are homoscedasticity. The heteroscedasticity in the sample data of this study may be manifested as different variances of error terms in different countries (individuals). If heteroscedasticity is ignored, although the coefficient estimator remains unbiased and consistent, its standard error will be biased, which may lead to inaccurate confidence intervals. Therefore, we adopted a multi-level processing strategy: 1. Perform logarithmic transformation on the variables to alleviate skewed distribution and extreme values; 2. Using Cluster-Robust Standard Errors to deal with any form of intra-group correlation, that is, allowing for any form of correlation and heteroscedasticity among error terms within individuals (different years in the same country). Therefore, to more accurately measure the relative changes between variables and mitigate potential heteroscedasticity, logarithmic transformation was applied in this study. In addition, logarithmic calculation was not carried out for negative values. In model (1), ln $PS_{it}$ is the explained variable, indicating the degree of political stability of the country i in the year t; $DG_{it}$ represents the digital governance level of the country i in the year t; $\varnothing_i$ is the individual fixed effect respectively; $\varepsilon$ it is the random disturbance term; $\beta_0$ represents the intercept term of the model; $\beta_1$ represents the estimation coefficient of the independent variable; Control$_{it}$ represents a series of control variables, which include educational level (el), population size (ln ps) and the rule of law (ln rl).

**3.1.2 Intermediate effect model.** To verify H2 and examine whether the anti-corruption efforts mediates the effect of the digital government on political stability, model (1) is combined with the stepwise regression method. The intermediate effect models in this study are set as follows:

$$ln\ AC_{it} = \alpha_0 + \alpha_1 DG_{it} + \alpha_2 \sum control_{it} + \varnothing_i + \varepsilon_{it} \tag{2}$$

$$ln\ PS_{it} = \lambda_0 + \lambda_1 DG_{it} + \lambda_2 ln\ AC_{it} + \lambda_3 \sum control_{it} + \varnothing_i + \varepsilon_{it} \tag{3}$$

In model (2), ln $AC_{it}$ indicated the anti-corruption efforts of the country i in the year t; α1 represents the estimated coefficient of the digital governance level. In model (3), λ1 represents the estimated coefficient of the digital governance level; λ2 represents the estimated coefficient of anti-corruption. According to the analysis steps of the stepwise regression method, the first step is to regress the benchmark model (1). If β1 is significant, the digital governance level significantly

affects the degree of political stability. The second step is to regress model (2), and the intermediate variable is the anti-corruption efforts. If α1 is significant, then an increase in the digital governance level can enhance or reduce the anti-corruption efforts. In the third step, intermediate variables are introduced to the regression model (3). If λ1 is significant, the digital governance level can influence the degree of political stability in a country by influencing the strength of anti-corruption.

### 3.2 Variable selection and data description

**3.2.1. Explained variable.** This study focuses on the variable of political stability degree (ln PS). This study adopts Worldwide Governance Indicators (WGI), which was developed by the World Bank and is used to measure the comprehensive governance quality of various countries. The explained variable PS is derived from the "Political Stability and Absence of Violence/ Terrorism" indicator in WGI. It is worth mentioning that WGI is not only widely used in academic research, policy-making, and international comparison, but also provides an important reference for scholars to study governance issues, government policy-making, and international organizations to assess national development. However, the way WGI is measured is still being questioned. For example, indicators are too complex, there is a lack of time and space comparability, and there are hidden biases [41]. Undeniably, despite its limitations, WGI remains one of the authoritative indicator systems for measuring the state of global governance.

The development of WGI traces its origins to 1989 when the World Bank's report "Sub-Saharan Africa: From Crisis To Sustainable Growth" was published, the term "Crisis of governance" was first mentioned to describe the challenges facing the region [42]. Subsequently, the World Bank developed WGI based on its definition of governance: " ' the traditions and institutions by which authority in a country is exercised. ' " [43] The WGI incorporates existing data sources from more than 30 think tanks, international organizations, non-governmental organizations, and private companies around the world, and WGI aggregates these indicators into six governance dimensions covering more than 200 economies. The six dimensions of governance include: Voice and accountability (VA), political stability and absence of violence/ terrorism (PS), government effectiveness (GE), regulatory quality (RQ), rule of law (RL), and control of corruption (CC). According to Kaufmann and Kraay [44], the measurement standard of indicator PS is "capturing perceptions and views of the likelihood that the government will be destabilized or overthrown by unconstitutional or violent means, including politically motivated violence and terrorism." More precisely, The numerical calculation steps of each WGI's indicator are as follows: (1) Fully absorb the information of each data source – complete the matching of the six indicators with other independent data sources; (2) Collate and standardize each source data, and control the value range between 0 and 1; (3) Adjust the source data using Unobserved components model (UCM) to ensure that the composite index value of each governance indicator is between −2.5 and 2.5. The closer the value is to 2.5, the higher the quality of the governance indicator is. On the contrary, it indicates that the quality of the governance indicator is lower.

**3.2.2. Core explanatory variable.** The principal explanatory variable examined in this study is the digital governance level (DG). Government website construction, as the foundation and key form of expression in the field of digital governance, can significantly improve the working efficiency of the government and effectively improve the public image of the government [45,46]. It can be said that e-government is the key carrier of digital governance a significant part of digital governance, which can reflect the maturity of government digital services. Moreover, due to the differences in the degree, frequency, and way citizens use government websites, citizens' perception of government transparency will vary, which will further affect their evaluation of government service capabilities and their political attitudes [47,48]. Thus, it can be seen that the Internet usage rate, as the fundamental support of digital governance, is an important condition for citizens to participate in government digital governance. Therefore, we combined the two indicators of e-government and Internet usage rate to construct a composite index to measure the level of digital governance.

The E-government indicator is derived from the Electronic Government Development Index (EDGI) constructed by the United Nations. EDGI is a comprehensive measurement index calculated by the weighted average of three standardized

indicators: Online Service Index (OSI), Telecommunication Infrastructure Index (TII), and Human Capital Index (HCI). Three indicators are calculated as follows: OSI is an evaluation of government online services of countries, which divides online services into four development stages: information release stage, interaction stage, transaction processing stage, and seamless integration stage. The score of each country is standardized according to the stage reached by the government's online services and the degree of completeness of the services, the value range is 0–1. TII collects the number of fixed telephone lines, the number of mobile phone subscriptions, the number of Internet users, broadband access speed, etc., normalizes the above data to eliminate the impact of dimensions, and assigns corresponding weights according to the importance of each indicator, then calculates the weighted average, and finally standardizes the score of each country, making its value range between 0 and 1. The HCI collects data on education indicators such as adult literacy rates, secondary mastery enrollment rates, and tertiary enrollment rates, as well as standardizing each country's final score on a range of 0–1 [2].

Internet usage rate is derived from the "Facts and Figures" report constructed by the International Telecommunication Union (ITU). ITU obtains data from a variety of sources, including telecom operators, telecom/ICT regulators and national ministries of member States, demographic and health surveys, ICT Research Surveys in Africa, Asian LIRNE Survey, etc., and uses the population of each country as a weight, then calculate a weighted average of the proportion of individuals using the Internet in each country, estimates of regional and global internet use are produced [49].

The indicator of digital governance is a composite index calculated by the weighted average of standardized e-government and Internet usage rate, which is calculated as follows:

$$DG = w_{eg}\frac{eg - \overline{eg}}{S_{eg}} + w_{iu}\frac{iu - \overline{iu}}{S_{iu}}$$

(4)

In model (4), DG represents the composite index – the digital governance level; weg and wiu are the weights of e-government and Internet usage rate, respectively; weg $\frac{eg-\overline{eg}}{S_{eg}}$ and wiu $\frac{iu-\overline{iu}}{S_{iu}}$ are standardized indicators of e-government and Internet usage.

**3.2.3. Mediating variable.** This study used the anti-corruption efforts (ln AC) as the mediating variable, which is selected from the Corruption Perception Index (CPI) conducted by Transparency International (TI). Since 2012, the CPI has expanded access to data, including data from 13 different surveys and assessments from 12 agencies. Each country must be evaluated from at least three different sources before it can be included in the CPI. The data obtained will be standardized with a range between 0 and 100, with 0 being highly corrupt and 100 being very clean. On closer inspection, a score greater than 80 indicates relative integrity; A score between 50 and 80 indicates some level of corruption; A score between 25 and 50 indicates high levels of corruption; A score below 25 indicates high levels of corruption. Therefore, Consequently, an increase in the CPI score signifies improved anti-corruption performance.

**Table 1. Summary of measurements.**

| Varibles | Measurment | Data Source |
|---|---|---|
| Dependent variable Political Stability (*In PS*) | Political Stability and Absence of Violence/ Terrorism: from approximately −2.5 (highly unstable) to 2.5 (highly stable) | World Bank |
| Independent variable Digital Governance (*DG*) | e-Government Development Index Facts and Figures (Normalized weighted average) | the United Nations International Telecommunication Union |
| Intermediate variable Anti-corruption Efforts (*In AC*) | Corruption Perception Index (CPI): from 0 (highly corrupt) to 100 (highly clean) | Transparency International |

**3.2.4. Other variables.** Table 1 shows other variables, which include educational level (el), population size (ln ps), and the rule of law (ln rl), which are represented as control variables. To accurately disentangle the influence of digital governance level on the degree of political stability, this study refers to existing literature and includes the consciousness, social, and institutional factors that may have a potential impact on the degree of political stability in the control variables. Among them, EL represented the consciousness factor, and the data was selected from the World Bank; PS represented the social factor, again using data from the World Bank; RL is the judicial aspect of the system, and the data was selected from the WGI published by the World Bank.

## 3.3 Data sources

Starting from the availability of data, this study excluded countries with severe data deficiencies to ensure the accuracy and reliability of the research. At the same time, considering the dynamic change process of the research, this study adopted cross-national balanced panel data from 112 countries from 2014 to 2023. Primary sources for this study include the World Bank, Transparency International, and the United Nations. Table 2 provides a statistical description of the variables as mentioned above.

## 4. Results and analysis

### 4.1 Descriptive statistics

Table 2 shows the descriptive statistics of the main variables. The mean value of the degree of political stability (ln PS) is 3.62, and the standard deviation is 0.779. Compared the standard deviation of ln PS with the mean value of ln PS, the fluctuation of the data is relatively small, indicating that although there are gaps in the degree of political stability among countries, the gaps are small. On the contrary, when combined with the minimum value of 1.194 and the maximum value of 4.586 of ln PS, it is found that the gap of the data is not that small, indicating that in the relatively concentrated distribution, there are still some countries whose degree of political stability is significantly different from the average level. The average digital governance level (DG) is 0.0004936, which is close to 0, indicating that the average digital governance level in countries is low. The standard deviation of DG is 0.978, which is larger than its mean value, and the data fluctuated greatly, indicating a large gap in the digital governance level among countries.

### 4.2 Benchmark regression

**4.2.1 The impact of digital governance on political stability.** This study used the individual fixed effect model to empirically test the relationship between the digital governance level and the degree of political stability. Specifically, this study used standardized weighted calculation of two indicators, which included e-government and Internet usage rate, to measure the digital governance level. Meanwhile, Political Stability and Absence of Violence/ Terrorism were to measure the degree of political stability. This study selected panel data from 112 countries during the period 2014–2023 to analyze the impact of digital governance level on the degree of political stability. Table 3 is the analysis result of benchmark regression. Column (1) lists the main explanatory variables used in this study to analyze how the digital governance level

**Table 2. Statistical description of variables.**

| Variable | Sample size | Average | The standard deviation | The minimum | The maximum |
|---|---|---|---|---|---|
| ln PS | 1120 | 3.62 | 0.779 | 1.194 | 4.586 |
| DG | 1120 | 0.0004936 | 0.978 | −2.023 | 1.382 |
| ln AC | 1120 | 3.758 | 0.417 | 2.944 | 4.489 |
| el | 1120 | 5.674 | 4.433 | 1.016 | 27.757 |
| ln ps | 1120 | 16.389 | 1.649 | 12.539 | 21.053 |
| ln rl | 1120 | 3.732 | 0.724 | 1.743 | 4.6 |

**Table 3. Benchmark regression results.**

| | (1) | (2) | (3) | (4) | (5) |
|---|---|---|---|---|---|
| | *ln PS* | *ln PS* | *ln PS* | *ln PS* | *ln PS* |
| *DG* | 0.031* | 0.057*** | 0.080*** | 0.046*** | 0.053** |
| | (0.018) | (0.018) | (0.019) | (0.018) | (0.023) |
| *cons* | 3.620*** | 7.426*** | 7.343*** | 5.080*** | 13.573*** |
| | (0.054) | (0.462) | (0.462) | (0.459) | (2.395) |
| Control variables | No control | control | control | control | control |
| Individual fixed effects | No | No | No | No | Yes |
| Time fixed effects | No | No | No | No | No |
| Sample size | 1120 | 1120 | 1120 | 1120 | 1120 |

Note: *, **and ***indicate that the estimated coefficients are significant at the level of 0.1, 0.05 and 0.01 respectively, the same as below.

affects the degree of government stability, and other factors are not considered in this column. Columns (2) to (4) list the estimated results after each control variable is introduced based on the results in column (1). Column (5) lists the results of introducing a single fixed effect based on the above results.

The estimated results in Table 3 showed that when the estimated coefficient of digital governance level is the only explanatory variable, the result is positively significant at the 10% level; When controlling variables are combined, the estimated coefficient of digital governance level was positively significant at 1% level; When combined with the individual fixed effect, the estimated coefficient of the digital governance level was positively significant at the 5% level, while the constant term is significant at the 1% level, indicating that the individual fixed effect captures part of the individual heterogeneity. Overall, the digital governance level was positively correlated with the degree of political stability, which verifies H1. In other words, as the digital governance level rises, the degree of political stability increases, which further supports the theoretical reasoning presented in this study. By the current international trend, strengthening the construction of digital infrastructure and improving the digital governance level will enhance the degree of political stability to a certain extent. At the same time, improving the digital governance level can improve public satisfaction by optimizing service supply, and it provides a foundation for countries to jointly address global challenges by deepening international cooperation. Therefore, accelerating the comprehensive and in-depth application of digital technologies, strengthening international digital cooperation, and narrowing the digital divide are international challenges that all countries should overcome in the future.

**4.2.2 Robustness test.** This part tested the robustness of the influence of digital governance level on the degree of political stability. Specifically, this study tested the robustness of benchmark regression by adjusting the sample period, and the test results are shown in Table 4. The original sample time range of this study was 2014–2023. We shortened the

**Table 4. Robustness test.**

| | (1) | (2) |
|---|---|---|
| | *ln PS* | *ln PS* |
| *DG* | 0.057** | 0.0053** |
| | (0.028) | (0.026) |
| *cons* | 18.570*** | 12.217*** |
| | (3.188) | (2.687) |
| Control variables | control | control |
| Individual fixed effects | Yes | Yes |
| Time fixed effects | No | No |
| Sample size | 896 | 784 |

original sample time range to 2015–2022, and the benchmark model was regressed again for the robustness test. Table 4 column (1) shows the regression result of the influence of digital governance level on the degree of political stability. The coefficient of the digital governance level is significantly positive at the 5% level, which is consistent with the benchmark regression result, and H1 has been verified.

Considering the changes in the global political stability situation in 2016, such as the US election, the turmoil in the Middle East, the political scandal in South Korea, the terrorist attacks in many places in Europe, and other events that have had an enormous impact on the degree of political stability, as well as the persistence of these events, we conducted a robustness test by eliminating samples. In detail, samples from 2016 to 2018 were removed, and the benchmark model was regressed again. The estimated results are shown in column (2) of Table 4. The influence of digital governance level on the degree of political stability is still positive and significant, which once again proves the robustness of the model.

**4.2.3 Heterogeneity analysis.** There is a gap in the level of economic development of different countries. Therefore, this study divided the sample into four groups based on the World Bank's income grouping criteria for economies in 2020 [50]. The four groups are: low-income economies ($1035), lower-middle-income economies ($1036 - $4045), upper-middle-income economies ($4046 - $12535), and high-income economies ($12536), and regression was performed for each group. The regression estimation results are listed in Table 5 columns (1) – (4). The results show that the effect of digital governance level on the degree of political stability is significant in lower-middle-income economies and high-income economies, but not in low-income economies and upper-middle-income economies. The reason for this outcome may be that lower-middle-income economies are in economic transition, and the usage of digital governance can bring about significant changes that affect political stability. At the same time, the digital governance level in high-income economies is more developed than in the other three economies. As a result, any adjustment could have a significant impact on the degree of political stability. In contrast, low-income economies have weak digital infrastructure and limited capacity for digital governance, so the effect on political stability is less. However, digital governance systems in upper-middle-income economies are relatively mature but have not reached a certain level, so the impact on the degree of political stability may level off.

The impact of digital governance level on the degree of political stability shows regional heterogeneity. As the research sample consists of 112 countries, which vary greatly in culture, system and development stage, it is difficult to capture their complex and multi-dimensional relationships by directly conducting full sample regression or simple regional dummy variable interaction terms. Therefore, this study adopted a stratified systematic sampling method to strategically reduce the initial sample. First, we stratify the overall sample based on two dimensions highly relevant to theoretical research: 1. Geographical region. According to the United Nations geoscheme [51], countries are divided into five continents (Asia,

**Table 5. Heterogeneity analysis of economic development levels.**

|  | (1) | (2) | (3) | (4) |
|---|---|---|---|---|
|  | Low-income economies | Lower-middle-income economies | Upper-middle-income economies | High-income economies |
| *DG* | −0.128 | 0.107* | −0.008 | −0.046* |
|  | (0.190) | (0.059) | (0.042) | (0.027) |
| *cons* | 13.746 | 10.791 | 3.230 | 7.087** |
|  | (10.523) | (6.919) | (6.280) | (3.150) |
| Control variables | control | control | control | control |
| Individual fixed effects | Yes | Yes | Yes | Yes |
| Time fixed effects | No | No | No | No |
| Sample size | 161 | 255 | 271 | 433 |
| R-squared | 0.877 | 0.924 | 0.863 | 0.951 |

Europe, Africa, America, and Oceania) to control the differences in macro historical and cultural backgrounds. 2. Digital governance level. Based on the existing data, the average value (0.0004932), minimum value (−2.023), and maximum value (1.382) of digital governance levels in various countries were obtained and segmented in an equal-width manner. Countries within each continent are classified into three levels: low digital governance level (within the range of [−2.023, −0.888]), medium digital governance level (within the range of [−0.888,0.247]), and high digital governance level (within the range of [0.247,1.382]). Secondly, within each "regionally Digital governance level" cell formed by the intersection of the above two dimensions (for example, "Asia - High Digital Governance Level"), we aim to extract as many equal numbers of countries as possible. This "equal division" principle ensures that our sub-samples are not dominated by a specific group (such as European countries with high digital governance), thereby guaranteeing that each level of digital governance is evenly represented across different regions. This is crucial for fairly testing regional differences. Thirdly, through the above procedures, we ultimately obtained a highly representative balanced panel sub-sample covering all regions and digital governance levels around the world, which is approximately one-third of the original sample. We admit that this sacrifices a portion of the sample size, but we believe that this trade-off brings a key advantage to the research, namely enhanced comparability. Because if only the full sample is used, the research finds that it might be dominated by specific regions with extremely high or low levels of digital governance and large sample sizes (such as Europe or Africa), while this method effectively 'matches' countries in different regions that are at similar stages of digital governance development. For instance, we can representatively compare "high digital governance countries in Africa" with "high digital governance countries in Asia", thereby more purely identifying the "regional" effect while controlling for the variable of "digital governance level".

The regression estimation results are listed in columns (1) to (3) of Table 6. The results show that the impact of digital governance level is "context-dependent" rather than universal. In column (1) of Table 6, the impact of digital governance level is negative but not significant. This indicates that in this group of countries, digital governance level has not effectively promoted stability and may even have a slightly negative impact by amplifying social conflicts or being abused. Another factor that also has a negative impact but is very significant is the education level, which conforms to the assumption of the "Malthusian trap" [52]: In the case of an improvement in the level of education but no improvement in the opportunity

**Table 6. Heterogeneity test of digital governance levels.**

|  | (1) | (2) | (3) |
|---|---|---|---|
|  | Low-level digital governance | Middle-level digital governance | High-level digital governance |
|  | *ln PS* | *ln PS* | *ln PS* |
| *DG* | −0.202 | 0.243*** | 0.028 |
|  | (−0.4064) | (2.5775) | (0.1902) |
| *el* | −0.024*** | 0.009 | −0.001 |
|  | (−4.6988) | (1.0501) | (−0.3346) |
| *ln ps* | 0.010 | −1.853** | −1.076* |
|  | (0.0073) | (−2.5318) | (−1.8266) |
| *ln rl* | 0.176 | 0.041 | 0.218 |
|  | (0.8602) | (0.2256) | (0.7199) |
| *cons* | 2.220 | 36.571*** | 21.710** |
|  | (0.0912) | (2.7312) | (2.0444) |
| Individual fixed effects | Yes | Yes | Yes |
| Time fixed effects | No | No | No |
| Sample size | 97 | 121 | 152 |
| R-squared | 0.916 | 0.928 | 0.952 |

structure, civic awakening may give rise to dissatisfaction with the current situation. However, the influence of the rule of law in this context is positive but not significant, indicating that in these countries, the rule of law itself may not be sound enough to serve as a stable cornerstone. Therefore, this group of countries may be confronted with deep-seated problems such as weak national capacity, incomplete systems or social divisions. The fundamental social and political issues (such as fairness, opportunity, and security) could be the primary contradictions among these countries. The "superstructure" such as digital governance is difficult to play an active role and may even exacerbate the contradictions due to improper use. For example, in this group of samples, Haiti's national conditions are consistent with this result. According to the EDGI report (2025), "As in 2022, Haiti remains at the lowest EGDI level in the region with ongoing political crises and conflicts severely undermining efforts to create a stable and effective digital infrastructure." In column (2) of Table 6, the impact of digital governance level on political stability degree is positive and significant. However, unlike the previous group of countries, the influence of the rule of law level in this group of countries is no longer significant. This does not mean that the rule of law is unimportant; rather, it indicates that the rule of law has become a common "infrastructure" in this group of countries, with a relatively small degree of variation. Therefore, the role of digital governance as an "efficiency enhancer" has been highlighted. For instance, the situation of Bangladesh in this sample is rather typical. The country is orderly moving on the path of sustainable development, integrating digital technology with people's livelihood and economic development, providing stable jobs, promoting industrial innovation and stabilizing infrastructure [53]. In column (3) of Table 6, the impact of digital governance level on national political stability is no longer significant. In other words, when all countries are mixed together to calculate the "average effect", the net effect of digital governance level disappears. This precisely proves that its positive effect column (2) and negative effect column (1) cancel each other out in different types of countries. This is also the most powerful evidence to prove the existence of regional differences. This result also confirms the "threshold effect". For digital governance to have a positive impact, it needs to cross a key "institutional threshold". The countries in column (1) are below this threshold. The main contradiction lies in fundamental institutional issues (such as the lack of effective rule of law and social injustice), and the effect of digital governance is not significant. The countries in column (2) have crossed this threshold and thus can reap the "digital dividend". Overall, whether the level of digital governance can enhance political stability depends on the institutional environment of the country.

### 4.3 Mediation mechanism test

**4.3.1 The mediating effect of anti-corruption efforts.** Based on the theoretical analysis above, the rise in the digital governance level can not only promote the improvement of the degree of political stability but also may be conducive to anti-corruption, and thus affect the degree of political stability. In other words, not only does the digital governance level have a direct impact on the degree of political stability, but the digital governance level also has an indirect impact on the degree of political stability by influencing the anti-corruption efforts. Therefore, this research studied the mediating role of the anti-corruption efforts, and the test results of this analysis are shown in Table 7.

According to the test procedure of the stepwise regression method, it is shown in column (1) of Table 7 that the estimated coefficient of digital governance level is positive and significant, indicating that with the improvement of digital governance level, the degree of political stability increases accordingly. Therefore, the total impact of the digital governance level on the degree of political stability is significant, confirming H1. Digital governance is a means for governments to use technology to change governance methods and improve government efficiency. Consequently, through digital governance, governments can use digital tools such as artificial intelligence, big data, etc., to optimize decision-making and improve decision-making efficiency. At the same time, from the perspective of national development, digital governance can promote the participation of social organizations, citizens, and enterprises in political life and enhance social cohesion. In terms of a country's security, digital governance can improve monitoring capabilities, strengthen early warning capabilities, and formulate effective measures in advance to avoid external attacks and internal disharmony. All of these effects of digital governance can enhance the degree of political stability.

 

**Table 7. Mediating effect test.**

| | (1) | (2) | (3) |
| --- | --- | --- | --- |
| | *ln PS* | *ln AC* | *ln PS* |
| DG | 0.053** | 0.017** | 0.057** |
| | (0.023) | (0.008) | (0.023) |
| ln AC | | | −0.207** |
| | | | (0.088) |
| cons | 13.573*** | 5.085*** | 14.624*** |
| | (2.395) | (0.856) | (2.431) |
| Control variables | control | control | control |
| Individual fixed effects | Yes | Yes | Yes |
| Time fixed effects | No | No | No |
| Sample size | 1120 | 1120 | 1120 |

Column (2) of Table 7 shows the impact of digital governance on the anti-corruption efforts. The result indicates that the higher the digital governance level in a country, the higher its integrity level. It can be said that digital governance can reduce corruption. The underlying logic is that digital governance through the use of information technology, enhances government transparency, optimizes government processes, breaks down information barriers among government departments, and promotes the development of the Internet and social media, thereby increasing information transparency, optimizing the operation of power, achieving data sharing, strengthening social supervision, and ultimately enhancing the intensity of anti-corruption efforts.

Column (3) of Table 7 shows the mediating effect of the anti-corruption efforts on the impact of the digital governance level on the degree of political stability. The anti-corruption efforts coefficient is positively significant, indicating that there is a mediating effect on the anti-corruption efforts, which verifies H2. To be precise, digital governance enhances the degree of political stability by raising the strength of anti-corruption. From an international perspective, corruption may undermine the international economic order [3]; From a national perspective, corruption can lead to economic instability in a country [54]; From a citizen's point of view, corruption can undermine political trust [27], and all of these hazards listed above can contribute to political instability. However, digital governance can reduce the degree of political instability by mitigating these hazards.

**4.3.2 Robustness test of mediating effect.** To test the robustness of the mediating effect of the anti-corruption efforts, this study conducted the Sobel test and Bootstrap test on the regression results respectively, and the test results are shown in Table 8. Column (1) in Table 8 is the Sobel test result, which indicates that the R-squared of the sober test is 0.631, and the imitative effect is relatively good. Moreover, the anti-corruption efforts coefficient is 0.379, and is significant at the 1% level, indicating that the anti-corruption efforts does have an intermediary effect and is robust.

The results of the Bootstrap test are listed in column (2) of Table 8. The results showed that Bootstrap 1 and Bootstrap 2 are significantly positive at the 1% level. Thus, digital governance can strengthen the degree of political stability by raising the strength of anti-corruption, once again verifying H2.

Although this study employed robust statistical tests such as Sobel and Bootstrap to identify mediating effects, a thorough discussion of the inherent limitations of the data itself is crucial for the correct interpretation of the research conclusions. First, the measurement of digital governance level faces inherent challenges. Many mainstream digital governance indices (such as the UN's EGDI) rely on expert surveys and business questionnaires. These data often reflect the "perception" of business executives or specific expert groups, rather than the actual usage experience of ordinary citizens or the objective effectiveness of the government's back-end operations. This "elite perspective" may lead to a systematic overestimation of the governance levels of countries that have invested more in digital propaganda but have insufficient

**Table 8. Robustness test of mediating effect.**

| | (1) | (2) |
| --- | --- | --- |
| | Sobel test | Bootstrap test |
| ln AC | 0.379*** | |
| | (0.079) | |
| Bootstrap 1 | | 0.030*** |
| | | (0.008) |
| Bootstrap 2 | | 0.216*** |
| | | (0.028) |
| cons | 3.640*** | |
| | (0.278) | |
| Control variables | control | control |
| Individual fixed effects | Yes | Yes |
| Time fixed effects | No | No |
| Sample size | 1120 | 1120 |
| R-squared | 0.631 | |

actual inclusiveness. Meanwhile, existing indices usually focus on the breadth of online service supply and digital infrastructure, but they do not adequately measure the "quality" dimension of digital governance. As a result, our research may capture the "efficiency" aspect of digital governance, but fail to fully capture its "fairness" or "rights" aspect, the latter of which may have a more complex relationship with political stability. Second, the degree of political stability encompasses various aspects. For instance, the "political stability and no violence/terrorism" indicator in the WGI used in this study. However, such measurements may focus more on "the stability of order" or "the stability of regime", and the capture of "the stability of the system" – that is, the long-term resilience and legitimacy of the political system itself – is relatively indirect. Therefore, our research finds that digital governance can promote stability, but the nature of this stability requires careful interpretation. The limitations of the above-mentioned data are not the fatal weakness of our research, but rather a common challenge faced by all cross-border quantitative political economy studies. They mainly affect the accuracy of our conclusion and the boundary conditions. Therefore, our conclusion should be understood as -- there is a connection between digital governance and political stability through a specific mechanism under the current mainstream measurement system.

## 5. Conclusion

### 5.1 Main conclusion

Digital governance offers the potential for countries to increase their degree of political stability. The main purpose of this study is to observe the mechanism of the influence of digital governance on the degree of political stability. Based on the panel data for 112 countries from 2014 to 2023, this study used fixed effect models and mediating effect models to empirically investigate the relationship between the digital governance level, the anti-corruption efforts, and the degree of political stability. At the same time, this study examined the heterogeneity of digital governance at different economic development levels.

First, digital governance exhibits a direct positive impact on the degree of political stability. An empirical analysis of panel data for 112 countries from 2014 to 2023 found that digital governance significantly enhanced political stability. This conclusion was tested by two robustness tests: adjusting the sample period and removing samples. Second, the impact of digital governance on the degree of political stability displays notable economic development heterogeneity. In lower-middle-income and high-income economies, digital governance has a significant impact on the degree of political stability,

but not in low-income and upper-middle-income economies. The logical explanation behind this conclusion is that a high digital governance level requires not only the economic base to maintain basic digital operations, but the economic base to upgrade the digital infrastructure and technology constantly as well. Third, digital governance has an indirect impact on the degree of political stability, and the anti-corruption efforts has a mediating effect to some degree. In other words, digital governance can enhance political stability by raising the strength of anti-corruption. Using Sobel and Bootstrap tests, this conclusion is robust and establishes the mediating role of the anti-corruption efforts.

## 5.2 Policy implications

These findings yield three concrete policy recommendations for strengthening political stability through digital governance.

First, establishing effective multilateral cooperation mechanisms for digital governance should be prioritized. Countries could jointly establish a framework for global digital governance, and discuss the formulation of fundamental principles and norms for global digital governance through multilateral platforms such as the United Nations and G20. At the same time, countries could call for strengthening technical assistance, led by developed countries, through technical assistance and financial support, to help low-income countries strengthen the construction of digital infrastructure, learn emerging technologies of digital governance, and enhance digital governance capacity.

Second, greater attention needs to be paid to transnational corruption challenges. Establish a global anti-corruption network through international organizations of anti-corruption, to intensify efforts to combat transnational corruption. In the meanwhile, bring in digital anti-corruption tools, such as big data, Blockchain, artificial intelligence, etc., to improve global corruption surveillance. Also, establish a global open anti-corruption network platform to increase public participation in anti-corruption globally.

Third, establishing robust mechanisms for global policy evaluation and feedback represents a critical imperative. To enhance policy responsiveness, it is imperative to develop a digital-enabled global policy evaluation platform that provides real-time monitoring of digital governance policy impacts and best practices, thereby enabling evidence-based policy adjustments across nations. Based on the policy implication mentioned ahead establish a global policy evaluation system. More importantly, the global policy evaluation platform could be used to collect feedback from the global public on digital governance policies, thereby improving the effectiveness of policies. Lastly, countries can work together to promote global policy innovation, optimize governance models, and enhance political stability.

Fourth, Countries at different digital governance level should implement different countermeasures. Firstly, for the groups in this study where the digital governance effect is not significant or even negative, the core challenges for these countries are the digital divide and the insufficiency of national basic capabilities. Therefore, these countries can prioritize investment in mobile network coverage, skip the large-scale deployment of expensive fixed broadband, focus on high-impact and low-complexity pilot projects, achieve rapid results, and accumulate public trust. Also, they could concentrate the limited investment in digital governance on the known areas with a high incidence of corruption, allowing citizens to see the improvements brought about by digital governance. Secondly, for the groups with significant digital governance effects in the results of this study, the goals of these countries should be "connection" and "empowerment". Specifically, these countries can break down the "data silos" among various departments through legislation or by establishing systems, and prioritize the formulation of data sharing standards. At the same time, they could establish a supervised testing environment for digital governance innovation. This can not only encourage innovation but also prevent new social conflicts caused by immature technology or design flaws, balancing efficiency and stability. Thirdly, for the countries with high levels of digital governance but insignificant influence in the results of this study, the challenge lies in the upgrading of technological ethics and governance paradigms. These countries should establish regulations requiring that the automated decision-making systems used by their governments must meet the requirements of transparency, interpretability and human supervision, and prevent "black box" algorithms. In conclusion, considering the differences in economic and digital governance levels among countries, an effective digital governance strategy must be diagnostic. For low-income

countries, it is a catalyst for development; For middle-income countries, it is a booster for institutional reform; For high-income countries, it is the guardian of social resilience. Decision-makers should first assess the stage and core challenges their country is at, and then select the policy toolkit that matches them for customized implementation.

### 5.3 Research limitations and future recommendations

While this research provides key conclusions and policy implications, there are limitations. Data selection is a prominent limitation of this research. For example, the Corruption Perception Index published by Transparency International as a measure of the anti-corruption efforts. The CPI, although currently the most widely used corruption indicator in the world, does not distinguish between different types of corruption, and it is mainly based on people's subjective perceptions, which, however, do not match the actual level of corruption exactly. in future studies, when scholars want to use the corruption perception index as the source of the mediating variable, they should combine the CPI with other assessment indicators (e.g., corruption convictions, asset recovery rates) to understand the situation more comprehensively. At the same time, subsequent research could build on our findings, conduct focused comparisons in specific regions, or adopt qualitative methods to reveal the subtle differences among various types of corruption. Moreover, future research can also add new explanatory variables, expand the scope of the research area, and find other possibilities to enhance political stability.

## Supporting information

**S1 Data. Minimal data.**
(XLSX)

## Author contributions

**Conceptualization:** Yaxing Zhao.

**Data curation:** Zhizhou Du.

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
