## [Decision Letter · Decision Letter 0]

4 Nov 2025

Dear Dr. Du,

Thank you for submitting your manuscript to PLOS ONE. After careful consideration, we feel that it has merit but does not fully meet PLOS ONE’s publication criteria as it currently stands. Therefore, we invite you to submit a revised version of the manuscript that addresses the points raised during the review process.

On the basis of reviewers' comments and my own reading, I have decided to give you the possibility to submit a revised version of the manuscript. Please go carefully through the suggestions coming from reviewers and submit an improved version of the manuscript. 

We look forward to receiving your revised manuscript.

Kind regards,

Massimo Finocchiaro Castro, PhD

Academic Editor

PLOS ONE

Journal Requirements:

“This research was funded by The Major Project of the National Social Science Foundation of China, "Research on Accelerating the Formation of Production Relations More Compatible with New Quality Productive Forces" (24ZDA021), and the Research on Biodiversity Security Monitoring and Early Warning in China under the Background of High-level Opening Up to the Outside World (22&ZD088).”

4. We note that your Data Availability Statement is currently as follows: All relevant data are within the manuscript and its Supporting with or without limitations.

6. We note you have included a table to which you do not refer in the text of your manuscript. Please ensure that you refer to Table 1 in your text; if accepted, production will need this reference to link the reader to the Table.

Reviewers' comments:

Reviewer's Responses to Questions

**Comments to the Author**

1. Is the manuscript technically sound, and do the data support the conclusions?

Reviewer #1: Yes

Reviewer #2: Yes

2. Has the statistical analysis been performed appropriately and rigorously?

Reviewer #1: Yes

Reviewer #2: Yes

3. Have the authors made all data underlying the findings in their manuscript fully available?

Reviewer #1: Yes

Reviewer #2: Yes

4. Is the manuscript presented in an intelligible fashion and written in standard English?

Reviewer #1: Yes

Reviewer #2: Yes

Reviewer #1: General Evaluation

1. Originality and Significance

• Strengths: The paper addresses a timely and relevant issue in the context of global governance, with a focus on digital governance's impact on political stability, mediated by anti-corruption efforts. It is significant because it provides empirical evidence using panel data from 112 countries between 2014 and 2023, which is a comprehensive dataset for this kind of analysis. The combination of digital governance, anti-corruption, and political stability is a well-rounded approach to understanding governance mechanisms on a global scale.

• Suggestions: While the study is innovative, it would benefit from a clearer distinction in how the digital governance strategies might differ in their impact across regions. The paper could explore more specific mechanisms or case studies that showcase regional variations or unique challenges to digital governance, beyond just the economic levels.

2. Methodology

• Strengths: The use of econometric models, such as the individual fixed effects model and the stepwise regression method for testing the mediation effects, is well-suited to address the research questions. The robustness checks and heterogeneity analysis (by income group) add strength to the empirical approach.

• Suggestions:

o The methodology section could provide a bit more detail regarding the assumptions of the models. For instance, while the paper mentions that logarithmic transformations were applied to some variables to address heteroscedasticity, further clarification on how this was addressed specifically in the models and why this method was chosen would be beneficial.

o The Sobel and Bootstrap tests for robustness are appropriate; however, it might be helpful to include a discussion of potential limitations or biases inherent in the data, such as the quality of governance data or discrepancies in the political stability index used.

3. Clarity and Structure

• Strengths: The paper is well-organized and the flow of information is logical. Sections like the introduction, literature review, and methodology clearly explain the theoretical foundation, research design, and data. The paper adheres to the general structure expected for academic papers in political science and economics.

• Suggestions: There are some minor language issues that need to be addressed for clarity. For example, terms like "political stability" could be explained further in the introduction for non-specialist readers, as well as more concrete examples of "digital governance" and "anti-corruption efforts" from real-world cases.

4. Data and Results

• Strengths: The results section is well-detailed, with multiple regressions and robustness tests supporting the hypotheses. The analysis of heterogeneity across different income levels is an important contribution, highlighting the differential impacts of digital governance across countries at different stages of economic development.

• Suggestions:

o The results could benefit from additional discussion on the limitations of the political stability index used. Although WGI is a widely accepted source, the potential biases in these data should be acknowledged.

o The study could also consider including additional variables, such as regional effects or a deeper analysis of corruption types, to give a more nuanced understanding of how digital governance interacts with political stability.

5. Conclusions and Policy Implications

• Strengths: The conclusions are relevant, offering clear policy recommendations based on the findings. These include establishing multilateral cooperation mechanisms for digital governance, combatting transnational corruption, and creating a global policy evaluation system, all of which are practical and actionable.

• Suggestions: The policy recommendations could be expanded to include more detailed actionable steps for specific countries or regions. For instance, low-income countries might need more tailored recommendations on digital governance capacity-building, considering their limited technological infrastructure.

Overall Assessment

• Strengths: The study is a valuable contribution to understanding the role of digital governance in enhancing political stability through anti-corruption efforts. The use of cross-national panel data makes the findings robust and globally applicable. The policy recommendations are practical and timely.

• Suggestions for Improvement:

o Expand on how digital governance might operate differently across various regions, particularly for countries at different income levels.

o Provide more detailed discussion on data limitations, particularly regarding the political stability index.

o The methodology could benefit from more transparency regarding model assumptions and choices.

The manuscript is generally well-written but could benefit from minor revisions for clarity, grammar, and sentence structure. It would be helpful to simplify some of the more complex sentences and correct any typographical errors to ensure the manuscript is fully intelligible.

The manuscript is generally well-written but could benefit from minor revisions for clarity, grammar, and sentence structure. It would be helpful to simplify some of the more complex sentences and correct any typographical errors to ensure the manuscript is fully intelligible.

Recommendation: Accept with revisions

The paper makes an important contribution to the literature and provides significant empirical insights. Major revisions, particularly in the clarity of methodology and the robustness of the data sources, would further strengthen the paper.

Reviewer #2: The manuscript is well-written and excellently articulated. The data is gathered in a good manner and then displayed as per standards. I think the work of the author is fully genuine and contributed to the field of knowledge. Only one suggestion; the author should clearly refer to methodology and findings in the abstract section.

**Do you want your identity to be public for this peer review?** For information about this choice, including consent withdrawal, please see our Privacy Policy

Reviewer #1: No

Reviewer #2: **Yes: ** Khurshaid

---

## [Author Response · Author response to Decision Letter 1]

12 Nov 2025

Dear Editor and Reviewers,

First and foremost, i would like to extend my sincere gratitude for the time and effort you have dedicated to reviewing my manuscript. Your invaluable feedback greatly aids in enhancing the quality of my research. Now, every comment was meticulously addressed, ensuring the the necessary adjustments were made to enhance the paper’s rigor and clarity. I believe that these revisions have significantly improved the manuscript, rendering it more comprehensive and refined.

Herein, I provide detailed responses to each of the comments made by the reviewers:

Comment 1:

Response: We greatly appreciate your suggestion and have made the appropriate revisions in the manuscript. We also show the revised content here.

<<<The impact of digital governance level on the degree of political stability shows regional heterogeneity. As the research sample consists of 112 countries, which vary greatly in culture, system and development stage, it is difficult to capture their complex and multi-dimensional relationships by directly conducting full sample regression or simple regional dummy variable interaction terms. Therefore, this study adopted a stratified systematic sampling method to strategically reduce the initial sample. First, we stratify the overall sample based on two dimensions highly relevant to theoretical research: 1. Geographical region. According to the United Nations geoscheme (2025), countries are divided into five continents (Asia, Europe, Africa, America, and Oceania) to control the differences in macro historical and cultural backgrounds. 2. Digital governance level. Based on the existing data, the average value (0.0004932), minimum value (-2.023), and maximum value (1.382) of digital governance levels in various countries were obtained and segmented in an equal-width manner. Countries within each continent are classified into three levels: low digital governance level (within the range of [-2.023, -0.888]), medium digital governance level (within the range of [-0.888,0.247]), and high digital governance level (within the range of [0.247,1.382]). Secondly, within each "regionally Digital governance level" cell formed by the intersection of the above two dimensions (for example, "Asia - High Digital Governance Level"), we aim to extract as many equal numbers of countries as possible. This "equal division" principle ensures that our sub-samples are not dominated by a specific group (such as European countries with high digital governance), thereby guaranteeing that each level of digital governance is evenly represented across different regions. This is crucial for fairly testing regional differences. Thirdly, through the above procedures, we ultimately obtained a highly representative balanced panel sub-sample covering all regions and digital governance levels around the world, which is approximately one-third of the original sample. We admit that this sacrifices a portion of the sample size, but we believe that this trade-off brings a key advantage to the research, namely enhanced comparability. Because if only the full sample is used, the research finds that it might be dominated by specific regions with extremely high or low levels of digital governance and large sample sizes (such as Europe or Africa), while this method effectively 'matches' countries in different regions that are at similar stages of digital governance development. For instance, we can representatively compare "high digital governance countries in Africa" with "high digital governance countries in Asia", thereby more purely identifying the "regional" effect while controlling for the variable of "digital governance level".

The regression estimation results are listed in columns (1) to (3) of Table 6. The results show that the impact of digital governance level is "context-dependent" rather than universal. In column (1) of Table 6, the impact of digital governance level is negative but not significant. This indicates that in this group of countries, digital governance levelhas not effectively promoted stability and may even have a slightly negative impact by amplifying social conflicts or being abused. Another factor that also has a negative impact but is very significant is the education level, which conforms to the assumption of the "Malthusian trap" (Grinin 2022) : In the case of an improvement in the level of education but no improvement in the opportunity structure, civic awakening may give rise to dissatisfaction with the current situation. However, the influence of the rule of law in this context is positive but not significant, indicating that in these countries, the rule of law itself may not be sound enough to serve as a stable cornerstone. Therefore, this group of countries may be confronted with deep-seated problems such as weak national capacity, incomplete systems or social divisions. The fundamental social and political issues (such as fairness, opportunity, and security) could be the primary contradictions among these countries. The "superstructure" such as digital governance is difficult to play an active role and may even exacerbate the contradictions due to improper use. For example, in this group of samples, Haiti's national conditions are consistent with this result. According to the EDGI report (2025), "As in 2022, Haiti remains at the lowest EGDI level in the region with ongoing political crises and conflicts severely undermining efforts to create a stable and effective digital infrastructure." In column (2) of Table 6, the impact of digital governance level on political stability degree is positive and significant. However, unlike the previous group of countries, the influence of the rule of law level in this group of countries is no longer significant. This does not mean that the rule of law is unimportant; rather, it indicates that the rule of law has become a common "infrastructure" in this group of countries, with a relatively small degree of variation. Therefore, the role of digital governance as an "efficiency enhancer" has been highlighted. For instance, the situation of Bangladesh in this sample is rather typical. The country is orderly moving on the path of sustainable development, integrating digital technology with people's livelihood and economic development, providing stable jobs, promoting industrial innovation and stabilizing infrastructure (ESCAP 2024). In column (3) of Table 6, the impact of digital governance level on national political stability is no longer significant. In other words, when all countries are mixed together to calculate the "average effect", the net effect of digital governance level disappears. This precisely proves that its positive effect column (2) and negative effect column (1) cancel each other out in different types of countries. This is also the most powerful evidence to prove the existence of regional differences. This result also confirms the "threshold effect". For digital governance to have a positive impact, it needs to cross a key "institutional threshold". The countries in column (1) are below this threshold. The main contradiction lies in fundamental institutional issues (such as the lack of effective rule of law and social injustice), and the effect of digital governance is not significant. The countries in column (2) have crossed this threshold and thus can reap the "digital dividend". Overall, whether the level of digital governance can enhance political stability depends on the institutional environment of the country.

Table 6. Heterogeneity test of digital governance levels

(1) (2) (3)

Low-level digital governance Middle-level digital governance High-level digital governance

ln PS lnPS lnPS

DG -0.202 0.243*** 0.028

(-0.4064) (2.5775) (0.1902)

el -0.024*** 0.009 -0.001

(-4.6988) (1.0501) (-0.3346)

ln ps 0.010 -1.853** -1.076*

(0.0073) (-2.5318) (-1.8266)

ln rl 0.176 0.041 0.218

(0.8602) (0.2256) (0.7199)

cons 2.220 36.571*** 21.710**

(0.0912) (2.7312) (2.0444)

Individual fixed effects Yes Yes Yes

Time fixed effects No No No

Sample size 97 121 152

R-squared 0.916 0.928 0.952

Comment 2:

Response: Your point is indeed pivotal, and we have adjusted the manuscript in line with your recommendation. We also show the revised content here.

Comment 2.1

<<<Although the individual fixed effects model effectively addresses a major source of omitted variable bias by eliminating individual characteristics that do not change over time, such as a country's historical culture and geographical environment, it cannot automatically guarantee that the error terms are homoscedasticity. The heteroscedasticity in the sample data of this study may be manifested as different variances of error terms in different countries (individuals). If heteroscedasticity is ignored, although the coefficient estimator remains unbiased and consistent, its standard error will be biased, which may lead to inaccurate confidence intervals. Therefore, we adopted a multi-level processing strategy: 1. Perform logarithmic transformation on the variables to alleviate skewed distribution and extreme values; 2. Using Cluster-Robust Standard Errors to deal with any form of intra-group correlation, that is, allowing for any form of correlation and heteroscedasticity among error terms within individuals (different years in the same country). >>>

Comment 2.2

<<<Although this study employed robust statistical tests such as Sobel and Bootstrap to identify mediating effects, a thorough discussion of the inherent limitations of the data itself is crucial for the correct interpretation of the research conclusions.>>> A more in-depth discussion is shown in the reply to Comment 4.1.

Comment 3:

Response: Your are correct in this regard. Upon re-evaluation, we have corrected the said oversight. We also show the revised content here.

Definition of political stability<<<According to Hurwitz (1973), political stability can be defined based on five aspects: (1) Stability refers to the absence of violence; (2) Stability refers to the durability of a government's lifespan; (3) The stability of its existence as a legitimate constitutional order; (4) Stability of the structure without change; (5) Stability as a multi-faceted social attribute.>>>

Examples of digital governance<<<Digital governance aims to utilize digital technologies and data resources to govern and manage various fields such as the economy, society, and government. Its application scope is constantly expanding, and an increasing number of countries are incorporating it into their governance frameworks. For instance, Uruguay has launched the firma.gub.uy website, offering advanced electronic signature services from multiple providers registered with the electronic certification department. This is convenient for individuals and enterprises to use and verify, aiming to support all online cross-border transactions, save users' time and money, simplify administrative processes, reduce transaction obstacles, and enhance the productivity and competitiveness of enterprises; also, Singapore's application of artificial intelligence in public services has enhanced service efficiency in the healthcare and transportation sectors (EGDI 2025). >>>

Cases of anticorruption efforts<<<For example, Russia has established competitive procurement procedures in electronic form on electronic platforms, allowing the use of artificial intelligence technology to detect economic anomalies and signs of formal competition in government contracts. Meanwhile, since January 1, 2021, cryptocurrencies have been regarded by many countries as objects of anti-corruption monitoring (Sergey 2020).>>>

Comment 4:

Response: We’ve considered your perspective and partially adopted your recommendation, implementing pertinent modifications in the paper. We also show the revised content here.

Comment 4.1

<<<Although this study employed robust statistical tests such as Sobel and Bootstrap to identify mediating effects, a thorough discussion of the inherent limitations of the data itself is crucial for the correct interpretation of the research conclusions. First, the measurement of digital governance level faces inherent challenges. Many mainstream digital governance indices (such as the UN's EGDI) rely on expert surveys and business questionnaires. These data often reflect the "perception" of business executives or specific expert groups, rather than the actual usage experience of ordinary citizens or the objective effectiveness of the government's back-end operations. This "elite perspective" may lead to a systematic overestimation of the governance levels of countries that have invested more in digital propaganda but have insufficient actual inclusiveness. Meanwhile, existing indices usually focus on the breadth of online service supply and digital infrastructure, but they do not adequately measure the "quality" dimension of digital governance. As a result, our research may capture the "efficiency" aspect of digital governance, but fail to fully capture its "fairness" or "rights" aspect, the latter of which may have a more complex relationship with political stability. Second, the degree of political stability encompasses various aspects. For instance, the "political stability and no violence/terrorism" indicator in the WGI used in this study. However, such measurements may focus more on "the stability of order" or "the stability of regime", and the capture of "the stability of the system" - that is, the long-term resilience and legitimacy of the political system itself - is relatively indirect. Therefore, our research finds that digital governance can promote stability, but the nature of this stability requires careful interpretation. The limitations of the above-mentioned data are not the fatal weakness of our research, but rather a common challenge faced by all cross-border quantitative political economy studies. They mainly affect the accuracy of our conclusion and the boundary conditions. Therefore, our conclusion should be understood as -- there is a connection between digital governance and political stability through a specific mechanism under the current mainstream measurement system.>>>

Comment 4.2

We appreciate the reviewer's valuable suggestion to delve deeper into variables like regional context or corruption typologies. We agree that these factors are intriguing and merit scholarly attention.

The primary focus of our current manuscript, however, is to establish the foundational, cross-national relationship between digital governance and political stability at a macro level. Introducing highly context-specific variables, while valuable, would shift the scope of our paper towards a comparative case study analysis, which is beyond its central aim. We were concerned that such an expansion might dilute the clarity and parsimony of our core theoretical model and empirical tests.

Instead, we have strengthened the Discussion section by incorporating the reviewer's idea as a key direction for future research. We now suggest that subsequent studies could build on our findings by conducting focused comparisons within specific regions or by employing qualitative methods to unpack the nuances of different corruption types.

Comment 5:

Response: We thank you profoundly for your detailed insights, which have been instrumental in refining our work. We also show the revised content here.

<<<Fourth, Countries at different digital governance level should implement different countermeasures. Firstly, for the groups in this study where the digital governance effect is not significant or even negative, the core challenges for these countries are the digital divide and the insufficiency of national basic capabilities. Therefore, these countries can prioritize investment in mobile network coverage, skip the large-scale deployment of expensive fixed broadband, focus on high-impact and low-complexity pilot projects, achieve rapid results, and accumulate public trust. Also, they could concentrate the limited investment in digital governance on the known areas with a high incidence of corruption, allowing citizens to see the improvements brought about by digital governance. Secondly, for the groups with significant digital governance effects in the results of th

---

## [Decision Letter · Decision Letter 1]

25 Nov 2025

Digital governance, anti-corruption and political stability: an empirical study using cross-national panel data

PONE-D-25-35315R1

Dear Dr. Du,

We’re pleased to inform you that your manuscript has been judged scientifically suitable for publication and will be formally accepted for publication once it meets all outstanding technical requirements.

Kind regards,

Massimo Finocchiaro Castro, PhD

Academic Editor

PLOS ONE

Additional Editor Comments (optional):

Reviewers' comments:

Reviewer's Responses to Questions

**Comments to the Author**

Reviewer #1: All comments have been addressed

2. Is the manuscript technically sound, and do the data support the conclusions?

Reviewer #1: Yes

3. Has the statistical analysis been performed appropriately and rigorously?

Reviewer #1: Yes

4. Have the authors made all data underlying the findings in their manuscript fully available?

Reviewer #1: Yes

5. Is the manuscript presented in an intelligible fashion and written in standard English?

Reviewer #1: Yes

Reviewer #1: Reviewer Letter Manuscript Title: Digital governance, anti-corruption and political stability: an empirical study using cross-national panel data

Thank you for the opportunity to review the revised version of this manuscript. I have carefully examined the authors’ response letter and the revised manuscript. Below, I provide an expanded, analytic evaluation assessing whether the authors have sufficiently addressed the concerns raised in the prior review round.

1. Originality, Significance, and Theoretical Framing

The revised manuscript demonstrates meaningful progress in addressing concerns regarding regional differentiation and the contextual mechanisms through which digital governance influences political stability. The addition of a two-dimensional stratification approach—based on UN regional groupings and digital governance levels—introduces a more nuanced analytical frame. The authors also provide a clearer theoretical justification for why digital governance may yield heterogeneous political outcomes depending on institutional maturity, bureaucratic capacity, and regional norms. These revisions significantly enhance the conceptual robustness of the study.

2. Methodology, Model Assumptions, and Analytical Transparency

The authors have strengthened the methodological rigor by clarifying model assumptions, addressing heteroscedasticity, and explaining the use of logarithmic transformations. The explicit justification for cluster-robust standard errors and the improved transparency in the description of the mediation analysis (including the Sobel and Bootstrap procedures) address previous concerns effectively. The analytical sequence is now well-articulated, offering a replicable and logically coherent empirical strategy.

3. Clarity of Key Concepts and Use of Illustrative Examples

The revised manuscript incorporates precise definitions of political stability, digital governance, and anti-corruption grounded in authoritative literature. The incorporation of concrete country examples—such as Uruguay, Singapore, and Russia—provides real-world grounding that enhances clarity and broadens the manuscript’s accessibility to readers not specialized in governance studies. This improves the pedagogical quality of the manuscript and strengthens its fit for a multidisciplinary journal such as PLOS ONE.

4. Data Quality, Measurement Limitations, and Scope of Variables

A major improvement in the revised version is the explicit acknowledgment of measurement limitations associated with the WGI political stability index and other governance indicators. The authors discuss potential biases, elite-driven valuation effects, and conceptual risks inherent in governance metrics. Their justification for not expanding the model to include corruption typologies or region-specific fixed effects is well-reasoned, based on the macro-comparative objective of the study. This addition demonstrates an appropriate degree of methodological humility and transparency.

5. Interpretation of Results and Depth of Policy Implications

The revised manuscript now offers a more detailed, stratified interpretation of empirical results across different income groups and digital governance environments. The policy section is substantially strengthened, providing differentiated recommendations tailored to the institutional and technological maturity of various country categories. These refinements improve the paper’s practical relevance and align the policy analysis more closely with the empirical findings.

6. Language Quality, Structure, and Overall Presentation

The authors have made noticeable improvements in language clarity, structural coherence, and conceptual flow. While minor editorial refinements may still be beneficial at the copyediting stage, the manuscript now reads clearly and professionally.

Overall Recommendation

The authors have fully addressed the major concerns raised in the previous review. The manuscript is now methodologically robust, conceptually clearer, and substantively more informative. I recommend acceptance with only minor editorial polishing.

**Do you want your identity to be public for this peer review?** For information about this choice, including consent withdrawal, please see our Privacy Policy

Reviewer #1: No

---

## [Editor Report · Acceptance letter]

PONE-D-25-35315R1

PLOS One

Dear Dr. Du,

I'm pleased to inform you that your manuscript has been deemed suitable for publication in PLOS One. Congratulations! Your manuscript is now being handed over to our production team.

Kind regards,

on behalf of

Prof. Massimo Finocchiaro Castro

Academic Editor

PLOS One